# Multiple interfaces between a serine recombinase and an enhancer control site-specific DNA inversion

Meghan M McLean[1][†], Yong Chang[1][†], Gautam Dhar[1][‡], John K Heiss[1], Reid C Johnson[1,2]*

[1]Department of Biological Chemistry, David Geffen School of Medicine, University of California, Los Angeles, Los Angeles, United States; [2]Molecular Biology Institute, University of California, Los Angeles, Los Angeles, United States

**Abstract** Serine recombinases are often tightly controlled by elaborate, topologically-defined, nucleoprotein complexes. Hin is a member of the DNA invertase subclass of serine recombinases that are regulated by a remote recombinational enhancer element containing two binding sites for the protein Fis. Two Hin dimers bound to specific recombination sites associate with the Fis-bound enhancer by DNA looping where they are remodeled into a synaptic tetramer competent for DNA chemistry and exchange. Here we show that the flexible beta-hairpin arms of the Fis dimers contact the DNA binding domain of one subunit of each Hin dimer. These contacts sandwich the Hin dimers to promote remodeling into the tetramer. A basic region on the Hin catalytic domain then contacts enhancer DNA to complete assembly of the active Hin tetramer. Our results reveal how the enhancer generates the recombination complex that specifies DNA inversion and regulates DNA exchange by the subunit rotation mechanism.

*For correspondence:
rcjohnson@mednet.ucla.edu

[†]These authors are joint first authors of this work

[‡]Present address: Department of Obstetrics and Gynecology, David Geffen School of Medicine, University of California, Los Angeles, United States

Competing interests: The authors declare that no competing interests exist.

## Introduction

Site-specific recombination reactions have evolved as a relatively simple solution to a myriad of biological problems including gene regulation, viral integration, DNA transposition, chromosome segregation, and the programmed creation of genetic diversity (*Craig, 2002*). Most site-specific recombinases can be classified into two structurally and mechanistically unrelated groups that are named for the use of either a serine or tyrosine as the active site residue (*Grindley et al., 2006*). Some reactions, such as those mediated by the tyrosine recombinases Cre and FLP, only require the recombinase and its cognate DNA binding sites, whereas others involve additional accessory proteins and assemble elaborate synaptic complexes that provide tight control over chemical and mechanical steps of the reaction.

The synaptic complexes formed by serine recombinases contain the two recombining DNA segments on the outside of a tetrameric protein core (*Dhar et al., 2004*; *Nollmann et al., 2004*; *Li et al., 2005*). All four DNA strands are cleaved by near simultaneous attack on the DNA backbone by the catalytic serines to form 5′-phosphoserine linkages, thereby generating double-strand breaks at both recombination sites. DNA strands are then exchanged by a subunit rotation mechanism where one pair of synapsed subunits, together with their covalently-bound DNA strands, rotates 180° relative to the other pair (*Stark et al., 1989*; *Dhar et al., 2004, 2009a, 2009b*; *Li et al., 2005*; *Bai et al., 2011*). Attack of the phosphoserine by the free 3′ OH ligates the DNA in the recombinant configuration.

Hin is a member of the DNA invertase subclass of serine recombinases that utilize a remote enhancer element to control early and late steps of the reaction (*Johnson, 2002*). Hin inverts a ~1 kb segment of chromosomal DNA between two 26 bp *hix* recombination sites in *Salmonella enterica* (*Zieg et al., 1977*; *Zieg and Simon, 1980*). Inversion switches the orientation of a promoter, resulting

**eLife digest** Many processes in biology rely on enzymes that break both the strands in a DNA molecule, then rearrange the strands, and finally join them back together in a new configuration. These recombination reactions can, for example, change the positions of genetic elements such as enhancers and promoters within the DNA molecule and, therefore, influence how a given gene is expressed as a protein. Cells need to be able to control recombination reactions because they can lead to leukemia and lymphomas if they go wrong.

The enzymes that catalyze these recombination reactions are called recombinases. One type of recombinase binds to specific sequences of DNA bases and uses an amino acid in the enzyme–usually serine or tyrosine–to break and rejoin the DNA strands. Recombination reactions require the assembly of complexes containing many proteins bound to DNA. Tyrosine recombinases form relatively simple protein-DNA complexes, and these have been studied in detail. Serine recombinases, on the other hand, form more elaborate protein-DNA complexes, and much less is known about these.

Now McLean et al. have unraveled the mechanism that a serine recombinase called Hin uses to reverse the direction of a stretch of chromosomal DNA in the bacteria *Salmonella enterica*. Inverting this stretch of DNA–which contains about 1000 base pairs–changes the position of a gene promoter that is responsible for the production of flagellin, which is the protein that enables the bacterium to move. This is one of the tricks that *Salmonella* uses to evade the immune system of its host.

Previous research has established that four Hin subunits and two copies of a protein called Fis are needed to invert this stretch of DNA: two Hin subunits bind to each of the two *hix* recombination sites, and the Fis proteins (which are dimers) bind to each end of an enhancer that is located between the *hix* sites. A protein called HU then causes the DNA to bend and form a loop, and the four Hin subunits and the two Fis dimers all come together at the enhancer to form a structure called the invertasome where the recombination reaction occurs. All four DNA strands at the crossover point are broken as a result of a near simultaneous attack by the catalytic serine amino acids in the Hin subunits. One pair of Hin subunits–and the two DNA strands attached to them–then rotate by 180 degrees around the other pair of Hin subunits. This means that the stretch of DNA between the *hix* sites is inverted when the DNA strands are rejoined at the end of the reaction.

Enhancers often regulate transcription and other reactions from a distance. McLean et al. reveal how an enhancer of a DNA recombination reaction works. The pairs of Hin subunits that initially bind to the DNA are not catalytically active, but when they are brought together by the enhancer and form a tetramer, they become active. Two of the Hin subunits are clamped onto the enhancer by the Fis dimers and by directly interacting with the enhancer DNA, but the other two (and the DNA strands attached to them) are free to rotate within the tetramer. In the *Salmonella* chromosome the enhancer is located close to one of the *hix* sites (~100 base pairs away from it), so the length of the DNA between the enhancer and *hix* site physically limits the number of Hin subunit rotations to just one.

in alternative expression of flagellin genes. Flagellar phase variation is one way that *Salmonella* evades the host immune system.

DNA inversion by Hin occurs upon assembly of the invertasome, a tripartite nucleoprotein complex made up of four Hin subunits bound to two *hix* sites and the recombinational enhancer element (*Figure 1A*) (*Heichman and Johnson, 1990*). The enhancer is a 65 bp DNA sequence that has recognition sites for the bacterial nucleoid-associated DNA binding and bending protein Fis on each end. Whereas the *hin* enhancer is normally positioned about 100 bp from one of the *hix* sites, it can efficiently activate DNA inversion many kb away from the closest *hix* site by DNA looping (*Johnson and Simon, 1985*; *Kahmann et al., 1985*). Hin dimers bound to each *hix* site are remodeled within the invertasome into a chemically-active tetramer that is also competent for subunit rotation. Crystal structure snapshots of several serine recombinases highlight the large conformational changes required for formation of the tetramer and reveal a flat and hydrophobic interface between rotating subunit pairs (*Yang and Steitz, 1995*; *Li et al., 2005*; *Kamtekar et al., 2006*; *Yuan et al., 2008*; *Keenholtz et al., 2011*; *Ritacco et al., 2013*).

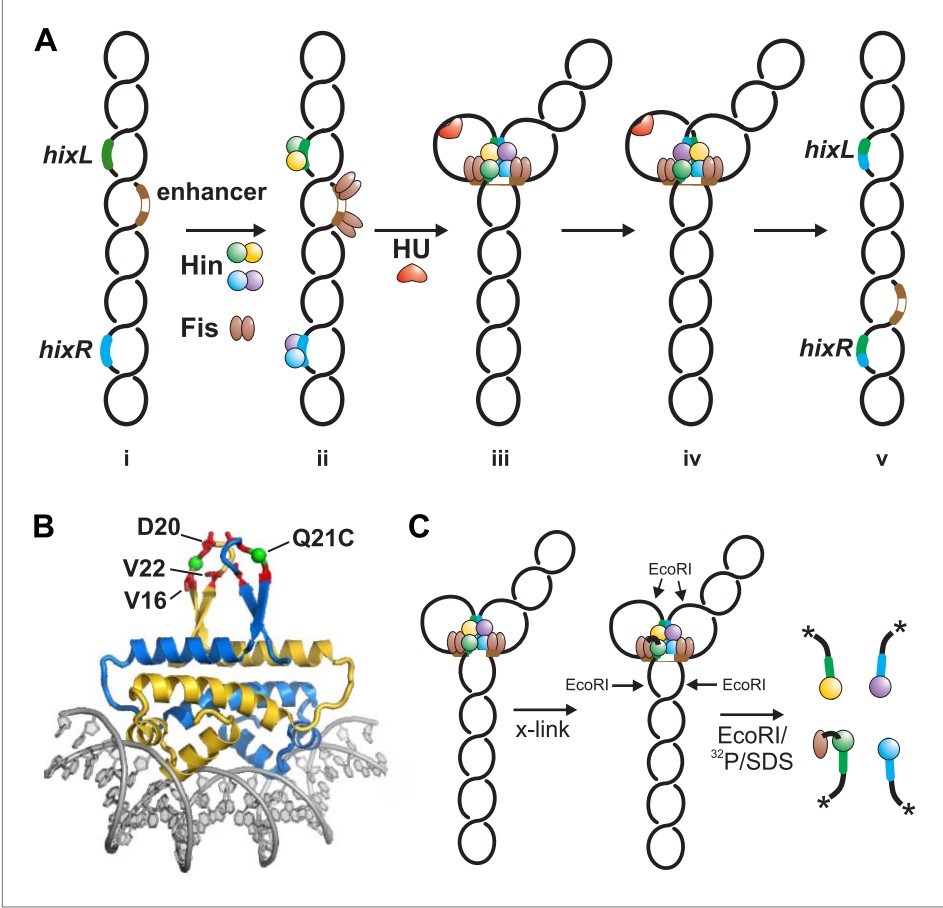

**Figure 1**. Hin-catalyzed DNA inversion reaction and outline of Fis-Hin crosslinking experiments. (**A**) The Hin reaction pathway proceeds by Hin dimers binding to *hixL* and *hixR* and two Fis dimers binding to the enhancer element (brown) (ii). The DNA bending protein HU aids in formation of the small (~100 bp) loop during assembly of the invertasome (iii). Each of the four Hin subunits cleaves the *hix* DNA by forming phosphoserine linkages with the 5' ends, and the DNA strands are exchanged by rotation of the purple and yellow subunits relative to the green and blue subunits (iv). The DNA is ligated by Hin in the recombinant orientation (v). (**B**) Fis-DNA crystal structure (PDB ID: 3IV5) highlighting the sole cysteine at the crosslinking target residue 21 (green spheres) in the protruding β-hairpin arms where surrounding residues that contact Hin are shown as red sticks. (**C**) Crosslinking is performed after incubating Fis and Hin under conditions stabilizing DNA-cleaved invertasomes. pRJ2372 (*Figure 1—figure supplement 1*) contains EcoRI sites flanking the *hix* sites such that each Hin subunit can be $^{32}$P-labeled by DNA polymerase and radiolabeled dNTPs after digestion with EcoR1, and the crosslinked Fis-Hin-($^{32}$P)DNA product is then detected by SDS-PAGE.

The following figure supplements are available for figure 1:

**Figure supplement 1**. Plasmid substrate design.

Formation of the supercoiling-dependent invertasome intermediate containing the Fis/enhancer element is the critical regulatory step in the Hin-catalyzed recombination reaction. It functions to ensure that intramolecular DNA inversion is the exclusive outcome of the reaction, promotes the Hin conformational changes required for DNA chemistry and exchange, and limits most subunit rotation reactions to a single 180° step (*Kanaar et al., 1990*; *Heichman et al., 1991*; *Moskowitz et al., 1991*; *Merickel and Johnson, 2004*) (*Figure 1A*). Whereas wild-type Hin is chemically inactive without the Fis/enhancer system, mutants that no longer require the Fis/enhancer element or DNA supercoiling have been isolated (*Klippel et al., 1988*; *Sanders and Johnson, 2004*; *Heiss et al., 2011*). These hyperactive mutants catalyze recombination promiscuously, being no longer restricted to promoting inversion between recombination sites on the same DNA molecule and no longer confined to a single round of subunit rotation.

Three residues on one of the mobile β-hairpin arms of the Fis dimer are required to activate Hin-catalyzed DNA inversion through direct contact with Hin (*Figure 1B*) (*Koch et al., 1991*; *Osuna et al., 1991*; *Safo et al., 1997*; *Dhar et al., 2009a*). In the present work we first identify residues within the DNA binding domain (DBD) of Hin that are contacted by the β-hairpin arms of Fis and identify which two of the four Hin subunits within the tetramer are associated with Fis in the active invertasome. We show that Fis contacts to inactive Hin dimers at an early step of invertasome formation lead to assembly of catalytically-active tetramers. Unexpectedly, we find that a localized basic surface on the Hin catalytic domain is also required for enhancer-dependent assembly of active tetramers and demonstrate using tethered chemical nucleases that it is associated with enhancer DNA between the two Fis dimers. These contacts enable us to construct a molecular model for the assembly of the invertasome structure at a plectonemic DNA branch that explains how the Fis/enhancer system controls orientation-specific synapsis, promotes the quaternary changes in Hin required for DNA cleavage and exchange, and prevents multiple subunit rotations.

## Results

### Fis contacts the DNA binding domain of Hin

We first used site-directed crosslinking approaches to identify regions on Hin that are required for association with the Fis/enhancer element. For these experiments, a cysteine (Q21C) was introduced into the tip of the Fis β-hairpin arms that are critical for activating Hin (*Figure 1B*). Invertasomes trapped in a DNA-cleaved state were assembled using Hin, Fis-Q21C, and a supercoiled plasmid substrate and subjected to crosslinking using the heterobifunctional agent AMAS (N-(α-maleimidoacetoxy) succinimide ester). AMAS chemically links Cys21 on Fis via the maleimide group to a lysine residue on Hin via the succinimidyl group. After quenching the crosslinking reaction, the *hix* recombination sites were separated from vector sequences using EcoRI, the EcoR1 ends were $^{32}$P-labeled, and the products were analyzed by SDS-PAGE (*Figure 1C*) (*Dhar et al., 2009a*). Crosslinking of Fis-Q21C to a lysine on Hin generates a radiolabeled Fis-Hin-DNA product, in addition to the covalently-linked Hin-DNA product from cleaved invertasomes (*Figure 2C*, Hin-wt panel; *Figure 2—figure supplement 1*). Because the two functional groups of AMAS are separated by only a 4.4 Å spacer, the crosslinked Hin lysine(s) are expected to be close to Cys21 on Fis. Experiments employing crosslinkers with different spacer lengths and with Fis-Q19C, where the cysteine on Fis appears less optimally positioned than at residue 21, are shown in *Figure 2—figure supplement 1*.

Hin contains 10 lysines per subunit; six are located in the catalytic domain and four in the DBD (*Figure 2A*). Two of the four lysines in the DBD are located in the unstructured C-terminal segment (*Chiu et al., 2002*), which is not present in other DNA invertases and can be deleted from Hin without affecting Fis-activated DNA inversion. The remaining eight lysines were each individually mutated to alanine and tested for crosslinking with Fis-Q21C; a truncation mutant missing the two C-terminal lysines was also evaluated. Each of the mutants accumulated substantial amounts of cleaved invertasomes at the time of crosslinking (10 min incubation), as also shown by the levels of covalent Hin-($^{32}$P-DNA) products (*Figure 2C*, *Figure 2—figure supplement 2*). Hin-K51A and K158A exhibited reduced amounts of Fis-Hin-($^{32}$P-DNA) crosslinked products relative to Hin-wt (*Figure 2B,C*, *Figure 2—figure supplement 2*). These results provide an initial indication that regions important for Hin association with the Fis/enhancer segment are located around Lys158 in the DBD and around Lys51 in the catalytic domain (*Figure 2A*).

To more directly probe the region on Hin that is positioned close to the Fis β-hairpin arms in the invertasome, cysteine-cysteine crosslinking between Fis-Q21C and Hin proteins containing cysteine substitutions at the eight native lysines was performed using BMOE (bis-maleimidoethane, 8 Å spacer). Hin-K146C at the N-terminus of helix 1 within the DBD generated ~15% crosslinked Fis-Hin products (*Figure 2B,D*), a level similar to that of wild-type Hin crosslinked to Fis-Q21C by AMAS. Hin-K158C at the C-terminus of the DBD helix 1 generated low levels of BMOE-mediated crosslinks with Fis-Q21C. No cysteine substitutions within the catalytic domain generated crosslinks (*Figure 2—figure supplement 3*), suggesting that the alanine mutations (e.g., Hin-K51A in *Figure 2B,C*) may have disrupted the AMAS-mediated crosslinking via an indirect mechanism (see below). Additional solvent-exposed residues within the DBD helix 1 region were converted to cysteine and those that generated substantial levels of DNA-cleaved invertasomes were tested for BMOE crosslinking with Fis-Q21C (*Figure 2D,E*). Hin-H147C, Hin-E150C, and Hin-Q151C within the α-helix formed crosslinks with Fis-Q21C, but H160C in the loop connecting to DBD helix 2 did not generate crosslinks. The 8 Å crosslinks between Fis-Q21C

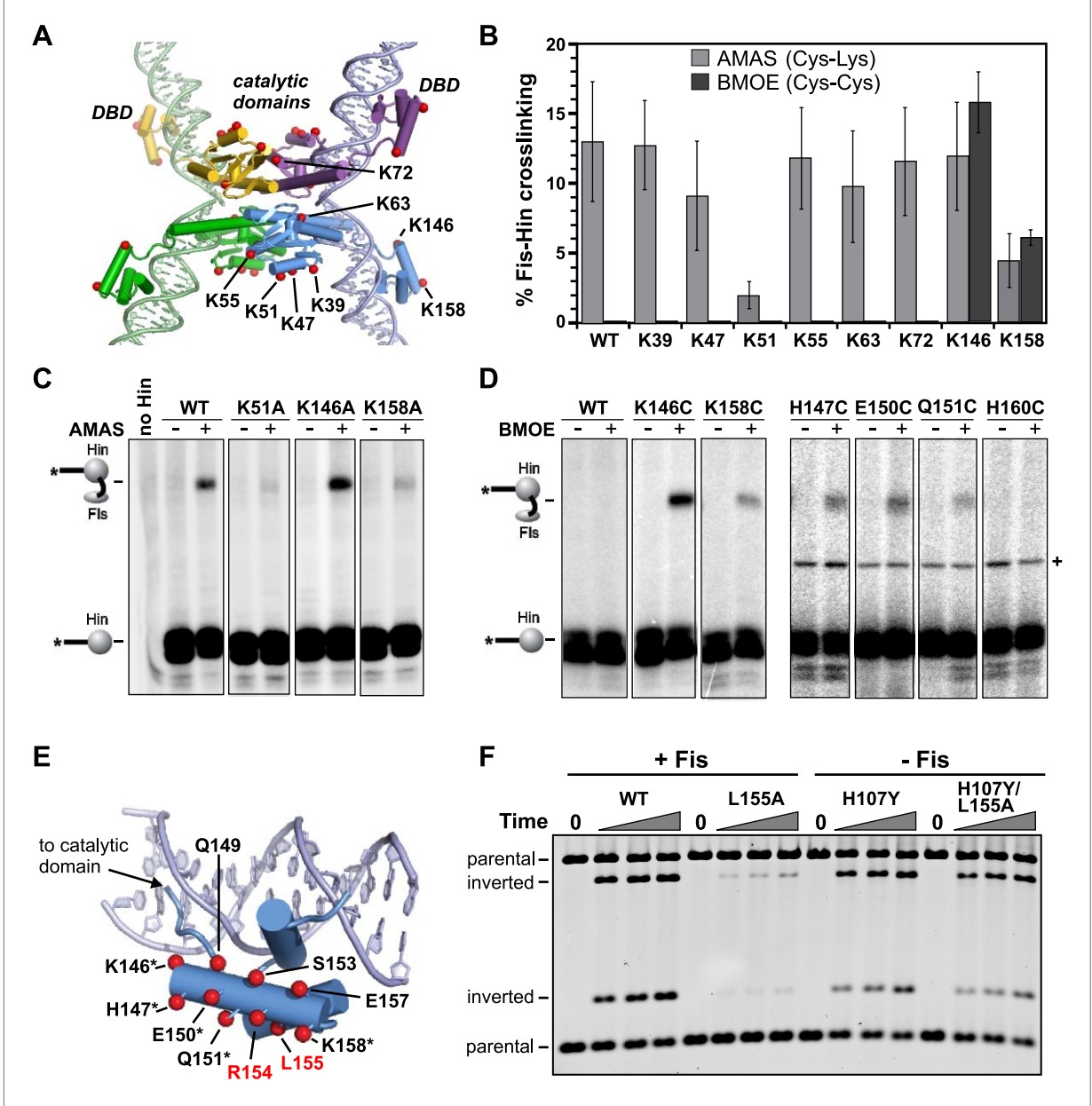

**Figure 2**. Identification of the Fis contact region on the Hin DBD. (**A**) DNA-cleaved tetramer model of Hin with the locations of eight lysines on each subunit shown as red spheres. (**B**) Fis-Hin AMAS or BMOE crosslinking efficiencies (mean and standard deviation from ≥three experiments) between Fis-Q21C and eight lysine to alanine (AMAS, 4.4 Å spacer, gray bars) or cysteine (BMOE, 8 Å spacer, dark bars) Hin mutants. Percent Fis-Hin crosslinks are relative to the Hin-($^{32}$P)DNA cleavage product. (**C** and **D**) Representative phosphorimages of AMAS or BMOE crosslinking experiments, respectively. A Hin- and crosslinker-independent contaminant band, which sometimes originates from an incompletely digested DNA fragment, is marked with a (+). (**E**) Hin DBD—*hixL* crystal structure (PDB ID: 1IJW) with residues evaluated for crosslinking or Fis-activated DNA inversion (*Table 1*) shown as spheres (Cβ atoms). Asterisks signify residues exhibiting BMOE crosslinking with Fis-Q21C when replaced with cysteine; red highlights the two key residues proposed to directly contact Fis. (**F**) DNA inversion activity of Hin-L155A in the absence or presence of the H107Y hyperactivating mutation enabling Fis/enhancer-independent inversion. Hin was added to reactions containing pMS551, HU, and Fis, as designated, and aliquots were taken at 0, 1, 2, and 5 min. Digestion with HindIII and PstI allows the inverted and parental DNA orientations to be distinguished. Hin-wt does not generate detectable inversions under these no-Fis conditions.

The following figure supplements are available for figure 2:

**Figure supplement 1**. Heterobifunctional crosslinking using different length crosslinkers targeting either Cys21 or Cys19 on Fis and a primary amine on Hin.

*Figure 2. Continued on next page*

*Figure 2. Continued*

**Figure supplement 2**. Complete set of AMAS crosslinking data between Cys21 on Fis and a primary amine on Hin.

**Figure supplement 3**. BMOE crosslinking data between Cys21 on Fis and cysteines introduced at eight lysine residues in Hin.

and five positions within helix 1 of the Hin DBD demonstrate that the Hin DBD is in close proximity to the Fis β-hairpin arms in the DNA-cleaved invertasome.

### Arg154 and Leu155 on the Hin DBD are critical for Fis-activated DNA inversion

An alanine scan of the DBD helix 1 was performed to determine the specific residues required for Fis/enhancer activation. Of the 11 solvent-exposed residues tested, Q151A, R154A, and L155A, exhibited strong reductions of Fis-activated DNA inversion in vivo and in vitro (*Table 1*). The most severe mutant, Hin-L155A, exhibits a >50-fold decrease of Fis-activated DNA inversion rates in vitro, comparable to the effects of the strongest Fis β-hairpin arm point mutations (*Safo et al., 1997*) (see also *Figure 7—figure supplement 1*). Although Hin-Q151A exhibits low activity, particularly in vitro, the role of Gln151 may be at least partially indirect because Hin-Q151C exhibits substantial Fis-activation and forms crosslinks with Fis-Q21C (*Figure 2D*) ('Discussion'). Additional changes were made at Hin-Arg154 and Hin-Leu155 and evaluated for DNA inversion (*Table 1*). The substitutions tested at Hin-Leu155 nearly inactivate recombination. Substitutions of Hin-Arg154 with polar residues (asparagine and serine) exhibit intermediate rates of DNA inversion, but alanine, cysteine, and aspartic acid are nearly inactive.

None of the above residues make DNA contacts in the Hin DBD co-crystal structures (*Feng et al., 1994*; *Chiu et al., 2002*) (*Figure 2E*), and mutants containing alanine substitutions at Arg154 or Leu155 bind *hix* DNA indistinguishably from Hin-wt (data not shown). In order to confirm that mutations of these residues specifically affect Fis-activation and not protein misfolding or other aspects of the recombination reaction such as site synapsis or DNA chemistry, rescue experiments were performed. Hin-R154A and Hin-L155A were each coupled to the gain-of-function mutation Hin-H107Y, which can catalyze recombination without the Fis/enhancer system (*Sanders and Johnson, 2004*; *Heiss et al., 2011*). As shown in *Figure 2F*, wild-type Hin recombines the plasmid substrate to a near equilibrium mixture of parental and inverted products within 5 min in the presence of Fis, whereas Hin-L155A inverts less than 2% of the substrate in the same amount of time. In contrast, both Hin-H107Y and the double mutant Hin-H107Y/L155A catalyze recombination in the absence of Fis at similar rates. Hin-R154A also efficiently promoted inversion when coupled with H107Y (data not shown). Taken together, we conclude that Fis specifically interacts with Hin residues Arg154 and Leu155 and that these contacts are required for DNA inversion catalyzed by wild-type Hin.

### Fis contacts the two Hin subunits at the base of the invertasome

We next asked which of the four Hin subunits were positioned proximal to the two Fis dimers in the invertasome. Site-directed crosslinking was performed on DNA substrates that enable a single Hin subunit to be labeled by having a restriction site flanking only one of the *hix* sites positioned appropriately for radiolabeling (*Figure 3A*, *Figure 1—figure supplement 1*). Crosslinking reactions were performed with Fis-Q21C and Hin-K146C using BMOE (8 Å). Fis-Hin crosslinks were only observed when Hin was bound to *hix1L* or to *hix2R*; no Fis-Hin crosslinks formed at *hix1R* or *hix2L*. Similar crosslinking experiments employing AMAS (cysteine-lysine) with wild-type Hin and Fis-Q21C also show that Fis only crosslinks with subunits bound to *hix1L* or *hix2R* (*Figure 3—figure supplement 1*). These crosslinking results demonstrate that Fis contacts the DBDs of only the bottom two Hin subunits within the cleaved invertasome as drawn in *Figure 3A*.

### Residues within the catalytic domain of Hin contribute to activation

As described above, Hin-K51A exhibits very low AMAS (Lys) crosslinking with Fis-Q21C, but crosslinking experiments with Hin-K51C provide no evidence for a location proximal to Fis (*Figure 2A–C*, *Figure 2—figure supplements 2 and 3*). Lys51 is located within the predicted helix-B of the Hin catalytic domain, which contains two other basic residues, Arg48 and Lys47 (*Figure 4A*). Hin-R48A failed to form detectable AMAS crosslinks under standard conditions (*Figure 2—figure supplement 2*). Hin-K47A associates

**Table 1.** Fis-activated DNA inversion activities of Hin DBD helix 1 mutants

| Mutant | DNA inversion in vivo* | DNA inversion in vitro† |
|---|---|---|
| WT | +++ | 0.31 ± 0.02 |
| WT (−Fis) | − | <0.001 |
| K146A | +++ | 0.30 ± 0.03 |
| H147A | +++ | 0.38 ± 0.02 |
| E148A | +++ | 0.19 ± 0.03 |
| Q149A | +++ | 0.37 ± 0.04 |
| E150A | +++ | 0.30 ± 0.05 |
| Q151A | ++ | 0.08 ± 0.01 |
| S153A | +++ | 0.17 ± 0.01 |
| R154A | + | 0.08 ± 0.02 |
| L155A | ± | <0.005 |
| E157A | +++ | 0.29 ± 0.03 |
| K158A | +++ | 0.17 ± 0.02 |
| R154S | + | 0.13 ± 0.03 |
| R154N | ++ | 0.17 ± 0.02 |
| R154C | + | 0.06 ± 0.01 |
| R154D | + | 0.02 ± 0.01 |
| L155V | ± | 0.03 ± 0.01 |
| L155G | ± | <0.01 |
| L155S | ± | <0.02 |
| L155T | ± | <0.01 |
| L155K | ± | <0.01 |
| L155R | ± | <0.02 |

*In vivo DNA inversion rates as measured by color development on lactose MacConkey media. +++ indicates red colonies developed within 24 hr, ++ red colonies between 25–29 hr, + red colonies between 30–34 hr, ± some red or papillations after 36 hr, and -no evidence of inversion (red) after 48 hr, as observed for no Hin or no Fis experiments.

†In vitro recombination rates (DNA inversions/ molecule/minute) obtained with purified proteins (mean and standard deviation from at least three determinations).

unstably with the enhancer as demonstrated by relatively efficient Fis-Hin crosslinking at early times but undetectable crosslinking after 20 min incubation (**Figure 4—figure supplement 1**). Fis-activated DNA inversion rates of Hin-R47A, R48A, and K51A are moderately decreased with R47A and K51A exhibiting about threefold reductions in vitro (**Figure 4B**). However, the combinations of K47A/R48A and R48A/K51A severely impair Fis-activated Hin inversion both in vivo and in vitro (**Figure 4B,C**). These mutations have no effect on Hin binding to *hix* (data not shown). To confirm that these residues are directly functioning in Fis/enhancer-activation, Hin-R48A/K51A was coupled to the gain-of-function mutation Hin-H107Y. Hin-R48A/K51A/H107Y promotes Fis-independent inversion at rates that are indistinguishable from the single H107Y mutant (**Figure 4C**), demonstrating that these mutations are not disturbing Hin catalytic properties. We conclude that basic residues in the Hin helix-B region function in the Fis/enhancer-dependent activation step of the Hin-catalyzed DNA inversion reaction.

## Hin helix-B mutations cause processive DNA exchanges

Fis/enhancer-association with the activated Hin tetramer inhibits multiple subunit rotations (processive recombination) during the DNA exchange reaction because the small (~100 bp) loop between *hixL* and the enhancer element prevents multiple windings of DNA (**Figure 4D**) (**Heichman et al., 1991**; **Merickel and Johnson, 2004**; **Dhar et al., 2009a**). Therefore, DNA exchange is normally limited to a single 180° rotation step, resulting in an unknotted inversion product. Additional subunit rotations, which generate DNA knots of increasing complexity, can occur when the enhancer is released from the invertasome complex. Reactions employing Fis mutants that have weakened interactions with Hin (**Merickel and Johnson, 2004**), or Hin DBD mutants that have weakened interactions with Fis (**Figure 4—figure supplement 2**) lead to processive recombination. Substrates containing long DNA segments between the enhancer and each *hix* site also exhibit increased processive recombination because multiple DNA windings in the context of large DNA loops do not restrict subunit rotation (**Heichman et al., 1991**; **Merickel and Johnson, 2004**; **Dhar et al., 2009a**).

Single-round knotting experiments were performed to test whether mutations within the Hin helix-B region increase processive DNA exchanges. Hin was incubated with Fis and the plasmid substrate for 5 min under conditions that accumulate DNA-cleaved invertasomes. A portion of the reaction was then quenched with SDS and evaluated for the amount of substrate reacted (**Figure 4—figure supplement 3A**). Under these conditions wild-type Hin cleaves 70–80% and helix-B mutants cleave 30–50% of the plasmid DNA. The remainder of the reaction was briefly switched to conditions allowing DNA ligation and the amount of knotted products were measured (**Figure 4E**, **Figure 4—figure supplement 3B**). Two substrates were used: pMS551, which contains the native (99 bp) spacing between

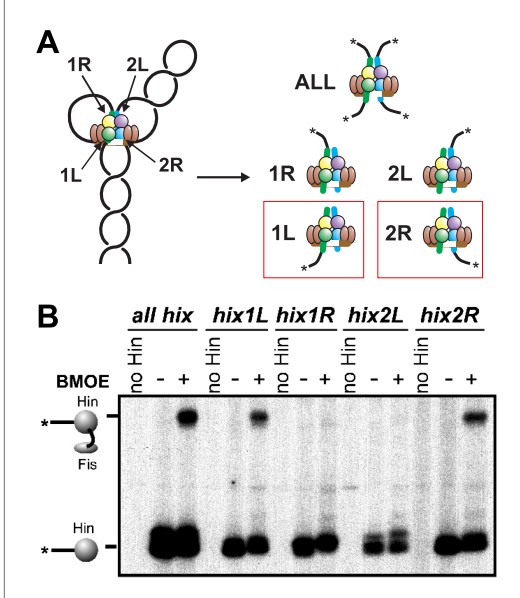

**Figure 3**. Fis contacts the two Hin subunits bound to half-sites *hix1L* and *hix2R* that are positioned at the base of the invertasome. (**A**) Plasmid substrates used to determine which Hin subunits contact Fis (see also *Figure 1—figure supplement 1*). The 3' DNA ends from each of the four Hin-DNA covalent complexes are labeled using the standard substrate pRJ2372 upon digestion with EcoRI and DNA polymerase fill-in (*all hix*). Four additional substrates were employed in which the DNA from only one Hin-DNA cleavage complex can be labeled due to the locations of the EcoR1 sites. DNA-labeled crosslinks between Fis-Q21C and the Hin protomers bound to the left half-site of *hix1* (*hix1L*) and the right half-site of *hix2* (*hix2R*) are observed (red outlines). (**B**) Fis-Hin crosslinking between Fis-Q21C and Hin-K146C on the different DNA substrates using BMOE.

The following figure supplements are available for figure 3:

**Figure supplement 1**. Heterobifunctional crosslinking (cysteine-lysine) between Fis-Q21C and Hin-wt was performed using N-succinimidyl iodoacetate (1.5 Å, no crosslinks obtained) or AMAS (4.4 Å).

the *hix1* site and the enhancer, and pMS634, which has a long (868 bp) spacer that does not restrict subunit rotation (*Figure 1—figure supplement 1*).

Only ~5% of the ligation products generated by Hin-wt on the short spacer substrate contained knots, reflecting a very low amount of processive DNA exchange even under reaction conditions where the invertasome is held in a DNA-cleaved structure for an extended time (*Figure 4E*). On the other hand, wild-type Hin knotted 30% of the reacted substrates with a long spacer between the *hix* site and the enhancer. Hin-R48A and Hin-K51A, however, are insensitive to the length of the loop between the *hix1* site and the enhancer; both mutants knotted about 40% of the reacted short spacer substrates corresponding to an eight-fold increase in processive DNA exchange compared with wild-type Hin. The increased processive DNA exchange by Hin-R48A and Hin-R51A provide further evidence that these mutations destabilize the association of the enhancer with the Hin synaptic complex.

## The Hin helix-B region is positioned close to enhancer DNA in the invertasome

To examine whether the Hin helix-B region is contacting enhancer DNA, the chemical nuclease FeBABE was covalently attached to cysteines introduced into the region (*Meares et al., 2003*). Invertasomes were assembled, and the enhancer region probed for DNA scission after activation of FeBABE with $H_2O_2$ and ascorbate (*Figure 5A*). Many of the tested cysteine mutants coupled to FeBABE exhibited low Hin-catalyzed DNA cleavage activity, but derivatives at residues 52–54, located near the C-terminal end of helix-B exhibited relatively high activity (*Figure 5—figure supplement 1A*). Hin-N54C-FeBABE, and to a lesser extent Hin-Y52C-FeBABE, generated two prominent scission sites within the center of the enhancer segment (*Figure 5—figure supplement 1B*). These are located between the Fis dimer binding sites, as demarked by DNA scission induced by Fis-N98C coupled with FeBABE

(*Figure 5—figure supplement 1B*); DNA scission by chemical nucleases coupled to Fis residue 98 has been used previously to map Fis binding site locations (*Pan et al., 1994*). Electrophoresis on DNA sequencing gels (*Figure 5B,C*) identified the precise locations of the scission sites by Hin-N54C-FeBABE and Fis-98C-FeBABE on the enhancer DNA sequence (*Figure 5D*). *Figure 5E* presents a molecular model showing the locations of the relevant Hin and Fis subunits on the enhancer DNA ('Discussion') together with the scission data.

## The DNA sequence between the Fis binding sites influences DNA inversion rates

We asked if the identity of the sequence between the Fis binding sites where the Hin helix-B region contacts in the invertasome is important for enhancer activity. When most of the DNA between the

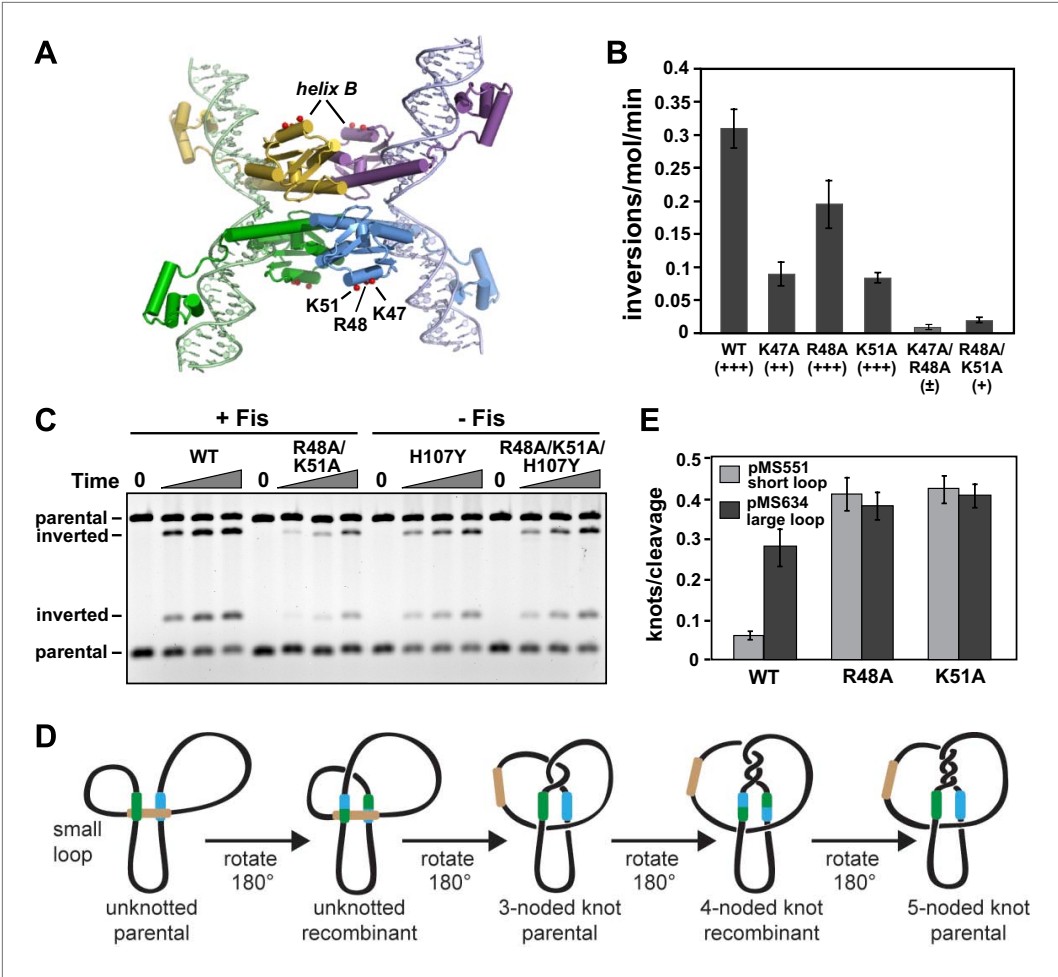

**Figure 4**. Residues within Hin helix-B function in Fis/enhancer-dependent Hin activation and control of subunit rotation. (**A**) DNA-cleaved Hin tetramer model with the locations of basic residues in the helix-B region highlighted with red spheres. (**B**) Fis-activated DNA inversion rates of helix-B Hin mutants reported as inversions/molecule/minute (mean and standard deviation from ≥three experiments). In vivo rates are given in parentheses; see *Table 1* for legend. (**C**) DNA inversion kinetics of a Hin helix-B double mutant. Inversion reactions were performed in the presence and absence of Fis for 0, 1, 2, and 5 min. Hin-R48A/K51A exhibits 10-fold slower kinetics than Hin-wt under Fis-activating conditions, but is fully competent for Fis-independent inversion when coupled to the hyperactivating mutation Hin-H107Y. (**D**) Schematic representation of topological changes during Hin recombination. Normally Hin ligates the DNA after a single DNA exchange by subunit rotation due to the small loop between the *hix* site and the enhancer, resulting in an unknotted inverted product. If the loop is large or the enhancer is released (as shown), multiple rounds of subunit rotations can occur resulting in DNA knots with increasing numbers of nodes. (**E**) Quantitation of DNA knotted forms relative to the amount of initially Hin-cleaved plasmid (mean and standard deviation from ≥three experiments) from single-round knotting experiments (see *Figure 4—figure supplement 3*). Hin-wt efficiently forms knots only on pMS634 containing the long spacer, whereas Hin-R48A and K51A are insensitive to spacer length, reflecting an unstable interaction with the enhancer.

The following figure supplements are available for figure 4:

**Figure supplement 1**. Hin helix-B mutation K47A destabilizes association of the Fis/enhancer with the active Hin tetramer.

**Figure supplement 2**. Partial disruption of Fis-Hin interactions leads to DNA knotting by processive subunit rotations.

**Figure supplement 3**. Primary data for single round knotting experiments summarized in *Figure 4E*.

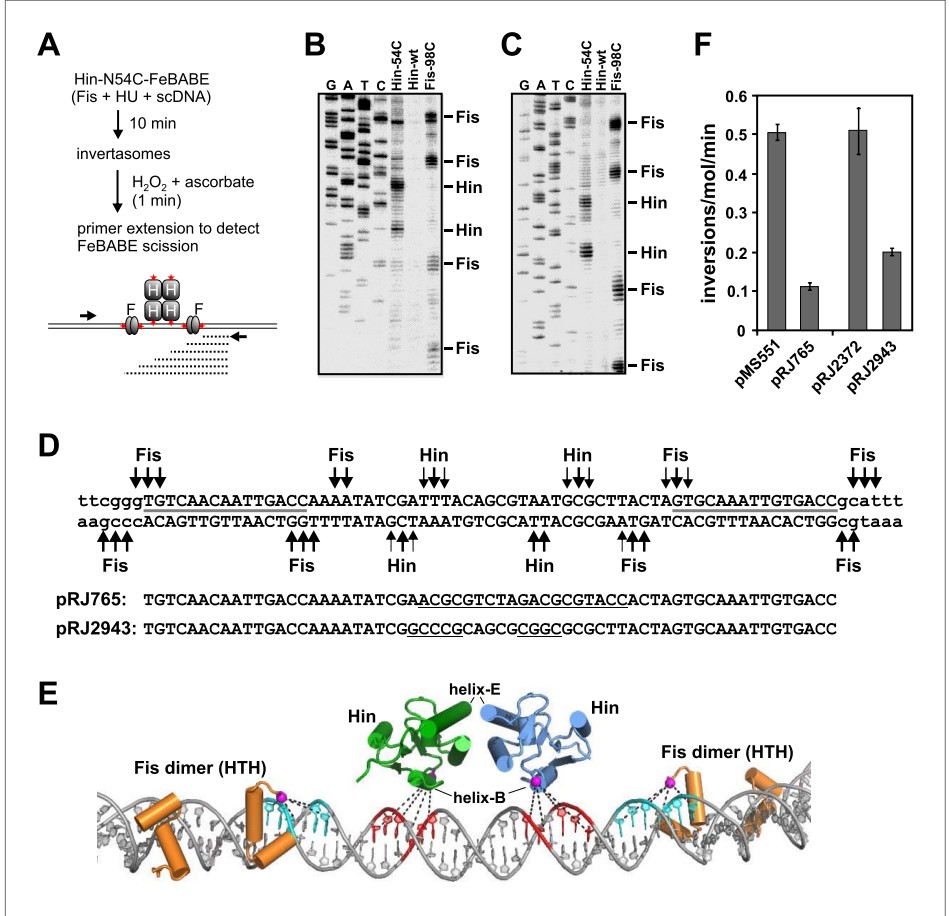

**Figure 5**. Localization of proteins on the *hin* enhancer DNA by site-specifically tethered FeBABE-mediated DNA scission. (**A**) Experimental approach for mapping enhancer DNA contacts by the Hin helix-B region. Red stars denote sites of FeBABE coupling. (**B** and **C**) Sequencing gels resolving primer extension products of DNA scission by FeBABE coupled to Hin-N54C, Hin-wt (control), and Fis-N98C next to dideoxy sequencing reactions. Top strand primer in **B**; bottom strand primer in **C**. (**D**) Sequence of enhancer DNA with primary Fis-N98C-FeBABE and Hin-N54C-FeBABE scission sites denoted. Thickness of arrows approximates scission efficiency based on several gels. Gray lines designate the 15 bp core Fis binding sites. Below are the sequences of two mutant enhancers used in panel **F**; underlined sequences are changes from the wild type. (**E**) Molecular model of enhancer segment within the invertasome structure ('Discussion', 'Material and methods', and *Figure 7C*). Shown are the helix-turn-helix regions of the Fis dimers (orange) and the catalytic domains of the two enhancer-proximal Hin subunits (green and blue). The Hin domains are rotated 40° about the y-axis relative to *Figure 2A*. Residues 98 on Fis and 54 on Hin have been replaced with cysteine and the Sγ atoms where FeBABE (12 Å to the Fe that generates hydroxyl radicals) is coupled are highlighted by magenta spheres. DNA scission sites generated by Hin-N54C-FeBABE are in red and by Fis-N98C-FeBABE are in cyan. (**F**) Inversion rates on mutant enhancers contained on pRJ765 and pRJ2943 relative to their wild-type enhancer parent substrates pMS551 and pRJ2372, respectively (*Figure 1—figure supplement 1*). Inversion rates are reduced by the mutations but are much greater than no-Fis reactions (<0.001 inversions/molecule/min).

The following figure supplements are available for figure 5:

**Figure supplement 1**. Activities and scission over the enhancer region by Hin mutants coupled to FeBABE.

two Fis binding sites is replaced by non-native sequence (pRJ765, *Figure 5D*), Hin inversion rates are reduced about fivefold (*Figure 5F*). When only the two A/T-rich segments identified to be in proximity to the Hin helix-B region by the FeBABE experiments are replaced by G/C-rich sequences (pRJ2943, *Figure 5D*), Hin inversion rates are reduced 2.5-fold (*Figure 5F*). We conclude that there is a modest effect of sequence identity within the enhancer DNA segments that are contacted by the helix-B region of Hin.

# Hin dimers crosslinked to Fis can transition into active tetramers

The experiment outlined in *Figure 6A* demonstrates that Fis productively interacts with Hin dimers at an early step to promote formation of chemically-active Hin tetramers. Disulfide-linked Hin-M101C dimers (*Figure 6B*) bind normally to *hix* sites but are locked in an inactive conformation (*Figure 6C*, lane 4) (*Haykinson et al., 1996*). Upon reduction of the disulfide bond and in the presence of Fis, Hin-M101C can proceed to generate DNA-cleaved invertasomes (*Figure 6C*, lane 5) and ligated inversion products (*Haykinson et al., 1996*). Fis-Q21C and disulfide-linked Hin-M101C dimers were incubated with the DNA substrate and subjected to Fis-Hin crosslinking for 15 s. The crosslinking reaction was quenched with DTT (plus free lysine), which also breaks the disulfide linkage and enables the reduced

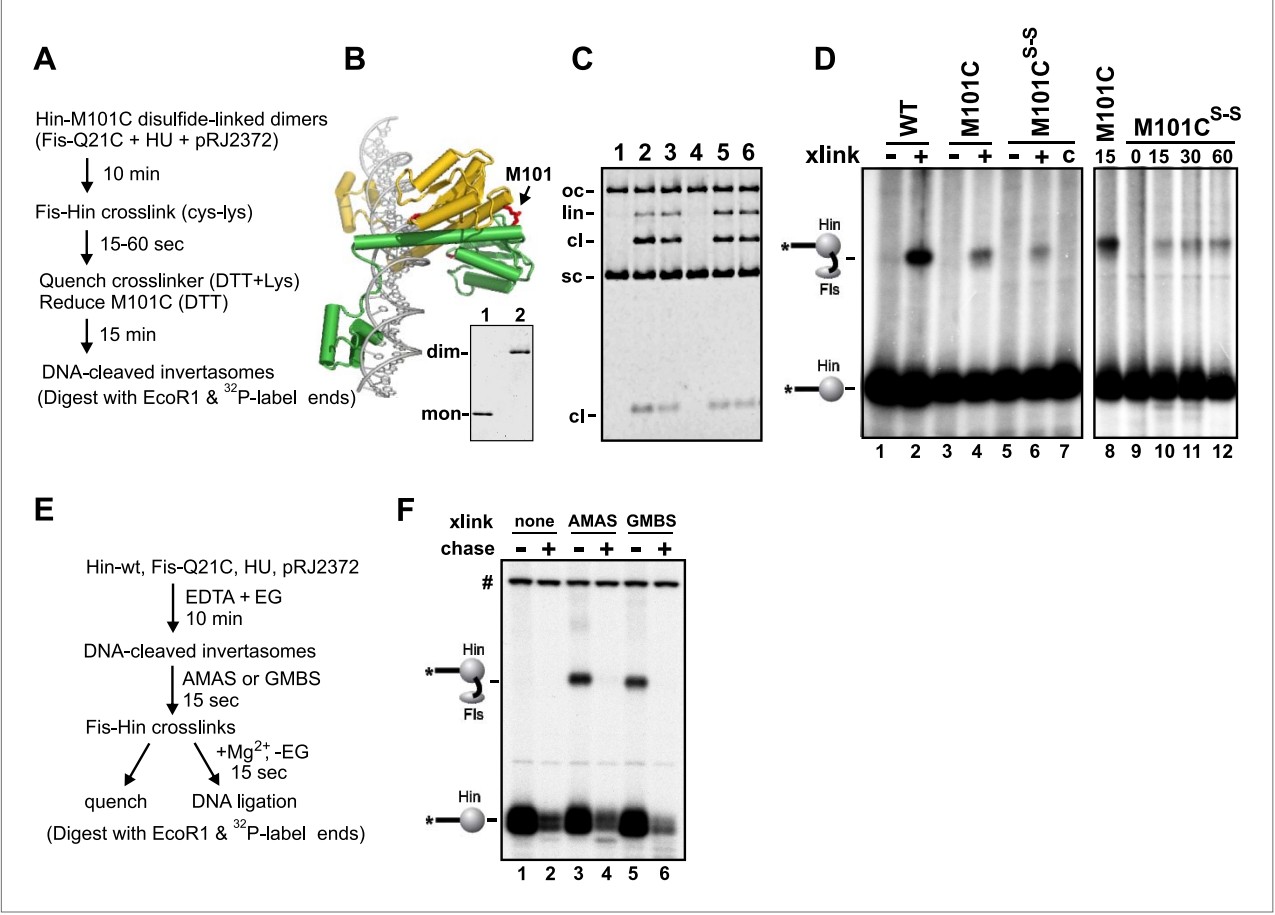

**Figure 6**. Fis-Hin connections at early and late steps in the DNA inversion reaction. (**A**–**D**) Hin dimers covalently linked to Fis proceed to tetramers active for DNA cleavage. (**A**) Outline of the experiment. (**B**) Hin dimer model highlighting Met101. Hin-M101C forms disulfide-linked dimers; insert shows non-reducing SDS-PAGE of reduced Hin-M101C (lane 1) and purified disulfide-linked dimeric Hin-M101C (lane 2). (**C**) Hin cleavage reactions with Fis-Q21C (10 min) displayed on an agarose gel. Lane 1, unreacted DNA; lane 2, Hin-wt reaction; lane 3, reduced Hin-M101C reaction; lane 4, disulfide-linked Hin-M101C[s–s] reaction; lane 5, disulfide-linked Hin-M101C[s–s] reaction then incubated 10 min with 10 mM DTT; lane 6, disulfide-linked Hin-M101C[s–s] reaction, crosslinked with GMBS (7.3 Å spacer), then incubated 10 min with 10 mM DTT, which inactivates the crosslinker and reduces the disulfide bond. (**D**) Fis-Hin crosslinking products displayed on an SDS gel. Lanes 1–6 are Hin-wt and reduced or disulfide-linked M101C crosslinked with Fis-Q21C for 30 s with GMBS as designated (+), the crosslinker was quenched, and the reaction incubated an additional 10 min under reducing conditions to form DNA-cleaved invertasomes. The presence of a Fis-Hin-([32]P)DNA crosslinked product in lane 6, as well as in lanes 10–12 where crosslinking times (s) were varied, demonstrates that covalently crosslinked Fis-Hin dimers can transition into Hin tetramers competent to cleave DNA. In lane 7, the DTT + lysine quench was added immediately prior to the crosslinker, demonstrating that the crosslinking in lanes 6 and 10–12 could not have occurred after reduction of the dimer. (**E** and **F**) DNA ligation by Hin proceeds when Hin is covalently crosslinked to Fis. (**E**) Outline of the experiment. (**F**) Control and crosslinked DNA cleavage reactions were either quenched directly (−) or chased by addition of 10 mM Mg[2+] and dilution of the ethylene glycol to induce DNA ligation. The amount of cleaved DNA remaining after switching to ligation conditions is assessed by the levels of Fis-Hin-([32]P)DNA or Hin-([32]P)DNA complex. Over 85% of the Hin-DNA and >95% of the Fis-crosslinked Hin-DNA covalent product is lost, demonstrating Fis-Hin association does not inhibit DNA ligation. The band designated # is a labeled DNA fragment from the substrate.

M101C dimers to then generate active tetramers and cleave the *hix* DNA. SDS-PAGE after EcoR1 digestion and radiolabeling of the EcoR1 ends revealed a substantial amount of Fis-Hin crosslinked products (*Figure 6D*, lane 6), albeit less than the amount generated with DNA-cleaved Hin-M101C$^{SH}$ invertasomes (*Figure 6D*, lane 4). Extended crosslinking times only slightly improved the yield of Fis-Hin dimer products (*Figure 6D*, lanes 10–12). Control experiments where DTT and lysine were added immediately before the crosslinker gave no detectable Fis-Hin crosslinked product (*Figure 6D*, lane 7), indicating that the Fis-Hin crosslinks could not have been formed after reduction of the disulfide-linked Hin-M101C dimer. We conclude that Hin dimers that are covalently crosslinked with Fis on the enhancer can be remodeled into catalytically-active tetramers.

### DNA ligation by Hin occurs efficiently when crosslinked to Fis

We tested whether Hin tetramers that are covalently crosslinked to Fis within DNA-cleaved invertasomes remain active for DNA ligation, the last chemical step of the reaction (*Figure 6E*). An earlier study with the related Gin DNA invertase concluded that the Fis-bound enhancer is normally released prior to the DNA ligation step (*Kanaar et al., 1990*). Stabilized DNA-cleaved invertasomes were assembled with Fis-Q21C and Hin-wt and crosslinked with AMAS (4.4 Å spacer) or GMBS (7.3 Å spacer). Uncrosslinked and Fis-Hin crosslinked invertasomes were then switched to reaction conditions allowing for ligation. As recognized by the loss of the Hin serine-DNA linkage in the Fis-Hin-($^{32}$P-DNA) and Hin-($^{32}$P-DNA) bands, both the Fis-Hin crosslinked (*Figure 6F*, lanes 3, 4 and 5, 6) and uncrosslinked (lanes 1, 2) complexes were able to efficiently promote DNA ligation. We conclude that ligation can efficiently occur when Hin is covalently linked to Fis, reinforcing earlier topological data indicating that the enhancer normally remains associated with the Hin complex throughout the reaction (*Heichman et al., 1991*; *Merickel and Johnson, 2004*; *Dhar et al., 2009a*).

## Discussion

We show that the Hin recombinase contacts the Fis/enhancer element at four distinct locations, which together promote assembly of the chemically-active Hin tetramer. These connections involve: (1) the DBDs of two Hin subunits with the β-hairpin arms of the Fis dimers, and (2) the catalytic domains of Hin with the enhancer DNA between the bound Fis dimers. As elaborated below, our new understanding of the connections between the eight protein polypeptides and three DNA segments, combined with existing molecular structures, enable us to construct a structural model of the Hin invertasome that catalyzes DNA exchange. We also provide experimental evidence for an early intermediate in the invertasome assembly pathway consisting of Hin dimers associated with Fis at the enhancer. The molecular architectures of intermediate and DNA-cleaved invertasome complexes reveal how the Fis/enhancer element functions at different steps to initiate and regulate DNA recombination by serine invertases.

### Assembly pathway and control of Hin recombination through the invertasome structure

The *hin* enhancer contains two Fis dimer binding sites separated by 47 bp between their centers (*Figure 5D*) (*Johnson and Simon, 1985*; *Bruist et al., 1987*; *Johnson et al., 1987*). A structural model of the Fis-bound enhancer ('Materials and methods') results in an S-shaped DNA structure due to Fis-induced bending (see *Figure 7F*; *Video 1*). Previous helical phasing experiments provide strong evidence that the shape of the enhancer segment is critical for function (*Johnson et al., 1987*). The Hin tetramer model, based on the crystal structures of the catalytic domain of the homologous serine recombinase γδ resolvase (PDB ID: 1ZR4) (*Li et al., 2005*) and Hin DBD (PDB ID: 1IJW) (*Chiu et al., 2002*), has been described previously (*Dhar et al., 2009a*). Extensive site-directed crosslinking data supports the validity of this model for the DNA-cleaved Hin synaptic complex, which is in a structure competent for DNA exchange by subunit rotation (*Li et al., 2005*; *Dhar et al., 2009a, 2009b*).

Manual rigid-body docking positioned the Hin tetramer model onto the Fis-bound enhancer such that the DNA strands at the base of the *hix* sites crossed the enhancer at the base of a plectonemic branch on (−) supercoiled DNA (shown schematically in *Figure 1A*), consistent with previous topological and electron microscopy studies (*Kanaar et al., 1988*; *Heichman and Johnson, 1990*; *Heichman et al., 1991*). The Fis-bound enhancer and Hin tetramer units fit remarkably well such that the critical amino acid triad on either of the flexible β-hairpin arms on both Fis dimers can be positioned proximal to Hin residues Gln151, Arg154, and Leu155 on the subunits bound to the *hix1L* and *hix2R* half sites (*Figure 7C*; *Video 1*). Previous experiments employing Fis heterodimers have shown that only one of

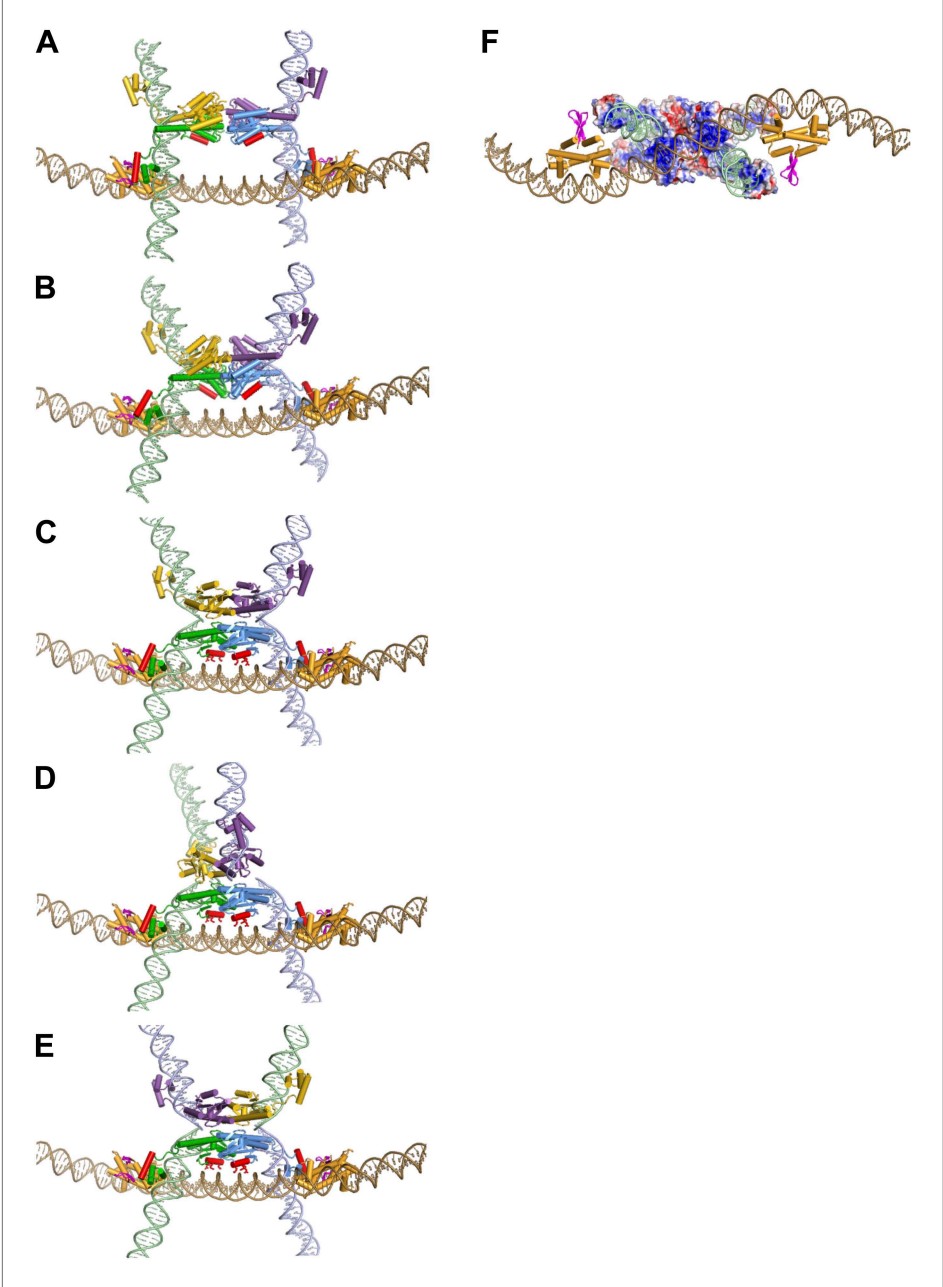

**Figure 7**. Assembly of the Hin invertasome. (**A**) Two Hin dimers bound to *hix* sites docked onto the enhancer (brown) with helix-B of the catalytic domain and helix-1 of the DBD highlighted in red. Fis dimers are gold with their mobile β-hairpin arms colored magenta. The *hix* DNA segments cross the enhancer to form 2 (−) nodes, consistent with a branch on negatively supercoiled DNA. (**B**) The pre-cleaved Hin tetramer model docked onto the enhancer. (**C**) DNA-cleaved Hin tetramer model docked onto the enhancer. In this conformation, basic residues within helix-B of the Hin catalytic domain (Lys47, Arg48, and Lys51, displayed as sticks) are close to the enhancer DNA, and the flat interface enabling subunit rotation has formed. (**D**) Rotation (50°) of the yellow and purple subunits relative to the green and blue subunits that remain bound to the Fis/enhancer element. (**E**) Complete subunit exchange positions the cleaved DNA ends into the recombinant configuration. See *Videos 1 and 2*. (**F**) DNA-cleaved Hin tetramer model rotated 90° about the x-axis relative to panel **C** with the electrostatic surface potential (± 4 kT e$^{-1}$) displayed. Two distinct basic regions (blue) surrounding helix-B on the bottom two Hin subunits are positioned adjacent to the enhancer DNA.

The following figure supplements are available for figure 7:

*Figure 7. Continued on next page*

*Figure 7. Continued*

**Figure supplement 1**. Chemical properties of Fis residues proposed to contact the Hin DBD.

**Figure supplement 2**. Electrostatic surface potentials of Hin and γδ resolvase (± 4.0 kT e⁻¹ for all images).

the β-hairpin arms from each of the Fis dimers is sufficient to activate Hin inversion (*Merickel et al., 1998*). Our data are consistent with the primary interaction surface between these proteins being composed of Val16 and Val22 on Fis and Leu155 on Hin together with Asp20 on Fis and Arg154 on Hin (*Figure 7—figure supplement 1*). Significantly, Hin residues Arg154, Leu155, and Gln151, which also may contribute to the Fis contact patch, are conserved among Fis/enhancer-dependent DNA invertases, but not the related resolvases (*Figure 8A*), further supporting the function of these residues in the Fis-activation step.

Our crosslinking and mutagenesis data unexpectedly identified an important region modeled to be on helix-B of the Hin catalytic domain that also plays a critical role in Fis/enhancer-activated DNA inversion. Like the helix-1 region of the DBD, the helix-B region is specifically required for Fis/enhancer activation because mutations within this region have no effect on Fis-independent recombination by hyperactive Hin mutants. This region forms a localized basic surface, which is positioned against the DNA segment connecting the two Fis binding sites of the enhancer in our model (*Figure 7C,F*, *Figure 7—figure supplement 2A,C*), as demonstrated by scission by the chemical nuclease FeBABE coupled to residue 54 located adjacent to helix-B. Arginines and lysines from the helix-B region protrude towards AT-rich (ATTTA and TAATG) minor grooves in a manner reminiscent of histone-DNA interactions in nucleosomes (*Figure 7C*) (*Luger et al., 1997*). The increased electronegative potential of the AT-rich, narrowed, minor groove may enhance interactions by these basic residues on Hin (*Rohs et al., 2009*). Indeed, enhancer sequences that are G/C-rich rather than A/T-rich exhibit moderately reduced activation rates (*Figure 5F*). The mutations could also be causing small changes in DNA curvature of the enhancer that impact invertasome assembly.

The basic character of key residues within the helix-B region is also uniquely conserved among the DNA invertase subclass of serine recombinases (*Figure 8B*). Of the 15 DNA invertases we have aligned, residue 47 is always a lysine, and residues 48 and 51 are equally represented by lysine and arginine, suggesting they are not involved in base-specific interactions. Members of the resolvase or integrase subclasses of serine recombinases, which are not regulated by a Fis/enhancer system, tend to not have lysines and arginines at the same positions, and the electrostatic surface over this region in the resolvase tetramer is much less basic (*Figure 7—figure supplement 2*). However, as discussed further below, both γδ/Tn3 and Sin resolvases contain nearby regulatory residues that mediate critical protein-protein contacts during assembly of their respective synaptic complexes (*Hughes et al., 1990*; *Murley and Grindley, 1998*; *Burke et al., 2004*; *Mouw et al., 2008*; *Olorunniji et al., 2008*; *Rowland et al., 2009*).

The invertasome model depicted in *Figure 7C* represents a late complex in the activation pathway in which all four DNA strands are cleaved and poised for exchange by subunit rotation. However, the Fis/enhancer system initially functions much earlier in the pathway to promote synapsis and remodeling of Hin dimers into the active tetramer, as wild-type Hin is unable to form active tetramers without the Fis/enhancer element (*Dhar et al., 2004*; *Sanders and Johnson, 2004*). We demonstrate here that Fis is able to productively contact inactive Hin dimers and that Hin dimers covalently crosslinked to Fis can transition into an active tetramer competent for DNA cleavage (*Figure 6A–D*). In *Figure 7A* DNA-bound Hin dimer models are docked to the Fis dimers on the enhancer in a manner approximating the Fis-Hin subunit contacts in the invertasome model. The two Hin dimers are sandwiched between the Fis β-hairpin arms such that the catalytic domains are adjacent to each other and thus optimally positioned to isomerize into a tetramer. Remodeling into the active tetramer may occur

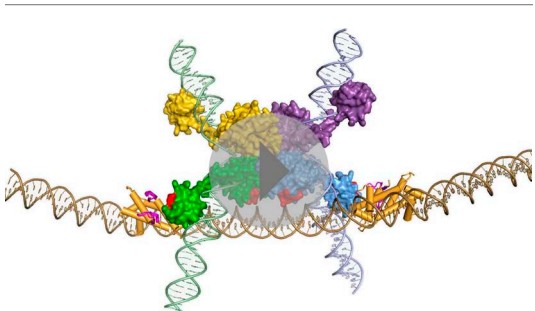

**Video 1**. Different views of the Hin invertasome model shown in *Figure 7C*.

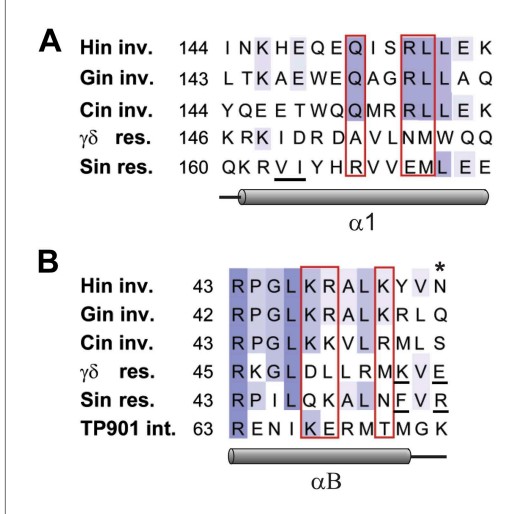

**Figure 8**. Sequence alignments of select serine recombinases over the regions that Hin contacts the Fis/enhancer. (**A**) Sequence alignment over the DBD helix-1 region of members of the Fis/enhancer-dependent DNA invertase family (Hin, Gin, Cin) and Fis/enhancer-independent resolvase family (γδ, Sin). Secondary structure is from Hin (PDB ID: 1IJW); the lengths of helix-1 in resolvases vary. Solvent-exposed residues Gln151, Arg154, and Leu155, which are proposed to directly or indirectly interact with Fis, are uniquely conserved among DNA invertases. Sin residues Val163 and Ile164 (underlined) mediate synapsis of regulatory subunits during formation of the Sin synaptic complex (***Mouw et al., 2008***). (**B**) Sequence alignment over the helix-B region of DNA invertases (Hin, Gin, Cin), resolvases (γδ, Sin), and a large serine recombinase (TP901 integrase). Secondary structure is from the Hin model; the helix-B boundaries are similar in most resolvase and Sin structures. Hin residue Asn54, the site of FeBABE coupling, is marked with an asterisk. The basic character of Hin residues 47, 48, and 51 are uniquely conserved among DNA invertases. Mutations at the conserved residue Arg43, which functions in folding or catalysis in other serine recombinases (***Olorunniji and Stark, 2009***; ***Keenholz et al., 2011***), also inactivate Hin. γδ resolvase residues Lys54 and Glu56 and Sin residues Phe52 and Arg54 (underlined) participate in protein-protein interactions between regulatory and catalytic subunits in these resolution reactions (***Hughes et al., 1990***; ***Murley and Grindley, 1998***; ***Burke et al., 2004***; ***Mouw et al., 2008***; ***Olorunniji et al., 2008***; ***Rowland et al., 2009***).

through an intermediate captured by the crystal structures of TP901 integrase (PDB ID: 3BVP) and γδ resolvase (PDB ID: 2GM5). In ***Figure 7B*** we model this pre-activated Hin tetramer intermediate based on the TP901 integrase tetramer (***Yuan et al., 2008***; ***Heiss et al., 2011***). The quaternary changes accompanying the remodeling of the dimers into the pre-activated tetramer result in a compact structure that more readily fits between the Fis dimers on the enhancer, similar to the DNA-cleaved tetramer model (see also ***Video 2***).

During the modeled conformational changes from pre-activated to DNA-cleaved tetramer the position of the Hin DBDs relative to the Fis dimers undergo relatively small changes (***Figure 7B,C***; ***Video 2***). However, the position of Hin helix-B changes substantially relative to the enhancer. Initially, helix-B is at a ~50° angle with respect to the enhancer DNA with its C-terminal end (Lys51) near the phosphate backbone and the remainder of the helix oriented away (***Figure 7B***). Further movement of the catalytic domains driven by coulombic forces between the helix-B region and the enhancer DNA will clamp the basic helix-B region onto the enhancer DNA, locking the subunits into the active conformation competent for DNA cleavage and subunit rotation (***Figure 7C***). These DNA interactions by the catalytic domain of Hin subunits bound to *hix1L* and *hix2R* augment the relatively weak interactions between the DBDs of the same subunits and Fis to stabilize the active invertasome structure.

Our model for Hin invertasome assembly reveals how the Fis/enhancer element functions to regulate recombination at multiple levels. Initially, Fis-Hin interactions, together with the propensity of plec-tonemically supercoiled DNA to form branched DNA structures containing 2 (−) nodes, localize the two *hix*-bound Hin dimers at the enhancer. As Fis-Hin interactions have not been observed by standard solution assays, conformational energy from DNA supercoiling appears essential to promote synapsis. Formation of the active Hin tetramer could then be driven by mass action forces. For example, dynamic scissor-like movements between subunits of each dimer may transiently expose the hydrophobic surfaces of the apposing dimer interfaces to initiate remodeling into the tetramer. Conformational energy from the Fis/enhancer segment may also be harvested to drive tetramer

assembly forward. This energy could be transduced from the twist deficit present in the negatively supercoiled DNA and/or by the mobile Fis β-hairpin arms to facilitate compaction of the two dimers into the tetramer. Spring-like mechanisms contributing to assembly of active recombination complexes also have been proposed for the Tn10 transpososome (***Chalmers et al., 1998***). Finally, attractive electro-static forces between the helix-B region and the enhancer DNA promote the final conformational change into the active tetramer structure.

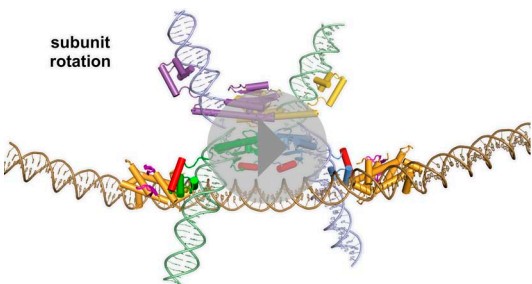

subunit rotation

**Video 2**. Assembly of the Hin invertasome and DNA exchange by subunit rotation within the invertasome structure (***Figure 7A–E***).

In our invertasome model the supercoiling energy in the looped DNA between the enhancer and *hix* sites drives the clockwise rotation of the top pair of subunits relative to the static bottom subunit pair that is fixed onto the enhancer (***Figure 7D,E***; ***Video 2***). A single clockwise rotation of subunits together with their linked DNA strands will result in inversion of the DNA between the *hix* sites with an accompanying loss of four supercoils, as shown for Hin- and Gin-mediated DNA inversion (***Kanaar et al., 1988***; ***Merickel and Johnson, 2004***). When the distance between a *hix* site and the enhancer is short, as in the native configuration in the *Salmonella* chromosome (~100 bp between *hixL* and the proximal Fis binding site), the small loop will inhibit additional subunit rotations because of torsional strain generated from multiple DNA windings (illustrated in ***Figure 4D***). However, additional rotations can occur under conditions where the loop is expanded by release of the Fis/enhancer from the Hin tetramer during the reaction or with substrates containing long segments of DNA between *hixL* and the enhancer. The structures of the resulting knotted DNA products are fully consistent with DNA exchange initiating within our invertasome model (***Kanaar et al., 1990***; ***Heichman et al., 1991***; ***Crisona et al., 1994***; ***Merickel and Johnson, 2004***). Thus, the multiple connections stabilizing the Hin tetramer onto the Fis/enhancer element function to restrict processive subunit exchanges. We experimentally demonstrate in this work that the final chemical step in the reaction, Hin-catalyzed DNA ligation, can efficiently occur when Fis and Hin remain physically connected (***Figure 6F***), implying that the invertasome normally remains intact over the course of the entire reaction.

## Summary and relationship with other serine recombinases

To summarize, we propose a multistep pathway for assembly of the recombinationally-competent Hin invertasome. Step 1 begins by the association of inactive Hin dimers bound to each of the *hix* sites with Fis dimers bound at the ends of the enhancer segment at the base of a supercoiled DNA branch (***Figure 7A***). In step 2 the localized Hin dimers are reconfigured into a pre-activated tetramer (***Figure 7B***). We propose this occurs via simultaneous opening of the dimer interfaces, which then transition into the tetramer in a process that may be facilitated by energetic forces transmitted by the Fis/enhancer segment. Formation of the initial tetrameric structure positions the C-terminal ends of helix-B towards enhancer DNA. In step 3, attractive electrostatic forces then 'pull' the basic helix-B region of the catalytic domains towards the enhancer DNA, clamping it against the enhancer segment between the two Fis dimers and completing the assembly of the tetramer that is competent for DNA cleavage and exchange (***Figure 7C–E***) (***Video 2***).

A hallmark of reactions promoted by the invertase/resolvase subfamily of serine recombinases is the formation of supercoiling-dependent, topologically-defined, higher-order synaptic complexes that are responsible for uniquely specifying the recombinant product. Whereas the invertasome structure utilizes a remote enhancer to align the recombination sites appropriately for inversion, resolvases assemble tightly interwrapped synaptosomes containing non-catalytic resolvase subunits bound to extended recombination sites, sometimes together with auxiliary DNA bending proteins, to align recombination sites for deletion (***Grindley et al., 2006***). Even though the molecular architectures of the complexes are very different, there are some striking similarities in the regulatory protein interfaces, which provide general insights into the control of this family of recombinases.

In the Sin-catalyzed deletion reaction, synapsis is initiated by an interaction between regulatory dimers bound to each recombination site via residues in their DBD helix 1 (***Figure 8A***) (***Mouw et al., 2008***). This interface can be related to the Fis-Hin interaction involving helix I of the Hin DBD that initiates invertasome formation. As with Hin, the DBD interaction by the Sin regulatory dimers is dispensable in the context of hyperactive mutants that promote recombination in the absence of the regulatory subsites. A second protein interface required for recombination by Sin, as well as the γδ/Tn3 resolvases, occurs between regulatory and catalytic subunits and can be related to the helix-B—enhancer DNA

interface in the Hin invertasome. For Sin this interaction involves residues Phe52 and Arg54 that are located immediately adjacent to helix-B (*Figure 8B*) (*Mouw et al., 2008*; *Rowland et al., 2009*). For γδ/Tn3 resolvases a cluster of residues (Arg2, Arg32, Lys54, and Glu56; *Figure 8B*) comprising the so-called 2–3′ interface between regulatory and catalytic dimers is important for assembly of the active synaptosome (*Hughes et al., 1990*; *Murley and Grindley, 1998*; *Burke et al., 2004*; *Olorunniji et al., 2008*). The surface of the γδ resolvase tetramer exhibits a basic patch that contains some of the 2–3′ interface residues but whose location is shifted from the basic region on Hin (see *Figure 7—figure supplement 2*). Like Hin, the interaction between regulatory and catalytic subunits in the resolvase reactions is dispensable in the context of strong hyperactive mutants that promote indiscriminant recombination (*Burke et al., 2004*; *Olorunniji et al., 2008*; *Rowland et al., 2009*). Taken together, the Hin, Sin, and γδ/Tn3 results imply that protein-protein or DNA-protein forces acting on this region of the catalytic domain may not only stabilize the active recombination complex but promote remodeling of this subfamily of serine recombinases into their active tetrameric conformation.

## Materials and methods

### Mutagenesis and purification of Hin and Fis mutants

Site-directed mutagenesis of the *hin* gene cloned into pET11a (*Merickel et al., 1998*) was performed using the QuikChange method. Native wild-type and mutant Hin preparations were obtained as described (*Heiss et al., 2011*). Homogeneous disulfide-linked Hin-M101C dimers (*Figure 6B*) were prepared by incubation with 10 mM oxidized DTT overnight at 4°C, followed by passage through a Thiopropyl Sepharose 6B column (GE Healthcare Life Sciences, Pittsburg, PA, USA). Fis purification was described in *Stella et al. (2010)*.

### Hin recombination assays

In vivo inversion rates were evaluated as described using inversion tester strain RJ3635 (*Heiss et al., 2011*), except that the strain also contained additional *lacI$^{qs}$*-D274N copies on a pACYC184-derived plasmid to reduce basal Hin expression. In vitro DNA cleavage and inversion reactions were performed as previously described (*Haykinson et al., 1996*) using pMS551 (unless otherwise stated), which contains the native *hixL*-enhancer spacing (*Figure 1—figure supplement 1*). For single-round knotting experiments, Hin and Fis were incubated with pMS551 and pMS634 under 30% ethylene glycol, Mg$^{2+}$-free conditions for 5 min to accumulate cleaved synaptic complexes, and an aliquot representing the DNA-cleaved sample was quenched with 1% SDS. The remainder of the reaction was then diluted ≥threefold in 37°C buffer containing no ethylene glycol and 10 mM MgCl$_2$ and incubated for 1 min to allow for DNA ligation. DNA knots were resolved in 0.84% agarose gels in Tris-phosphate-EDTA buffer after nicking with Nt.BsmA1 or DNase I in the presence of 200 μg/ml ethidium bromide. 'Knots/cleavage reaction' was calculated by dividing the percent knotted molecules by the percent cleaved molecules prior to ligation.

### Fis-Hin crosslinking

Fis-Hin crosslinking was performed essentially as described in *Dhar et al. (2009a)*. Typically, Hin and Fis were incubated with pRJ2372 under ethylene glycol, Mg$^{2+}$-free DNA-cleavage conditions for 10 min prior to crosslinking. Crosslinking with 0.4 mM AMAS (or GMBS) or BMOE (Pierce-Thermo Scientific, Rockford, IL, USA, dissolved at 10 mM in DMSO) was performed for 30 s and then quenched with 20 mM lysine pH 7.5/20 mM DTT/0.4% diethyl pyrocarbonate (DEPC) or 20 mM DTT/0.4% DEPC, respectively. After precipitation with ethanol, the DNA was digested with EcoR1 and BamH1 (BamH1 removes an interfering DNA band) and radiolabeled with α-$^{32}$P-ATP using Klenow. The products were then subjected to SDS-PAGE and phosphorimaging. Plasmid substrates were also used where a single Hin protomer is labeled after DNA cleavage by placing the EcoR1 site immediately adjacent to three of the four *hix* sites (see *Figure 1—figure supplement 1*). Placement of the EcoR1 site next to the *hix* site prevents radiolabeling by Klenow because of interference by the covalently-bound Hin (*Dhar et al., 2009a*).

### DNA scission by FeBABE-coupled proteins

Reduced Hin or Fis cysteine mutants (100 μg) were batch chromatographed on Heparin-Sepharose (GE Healthcare Life Sciences) to remove reducing agent (20 mM TCEP) and incubated with a 10-fold

excess of FeBABE (Pierce-Thermo Scientific or Dojindo Molecular Technologies, Inc., Rockville, MD, USA) overnight at 4°C in 20 mM HEPES (pH 7.5), 1 M NaCl, 0.1 mM EDTA, and 20% glycerol. Free FeBABE was removed by passage through a Zeba 7K MW spin column (Pierce-Thermo Scientific). The FeBABE-coupled proteins were added to standard Hin cleavage reactions, and DNA scission activated after 10 min incubation at 37°C by addition of 4 mM ascorbic acid for 5 s followed by 4 mM $H_2O_2$ for 30 s. After ethanol precipitation, DNA scission sites were detected by primer extension using 5′-$^{32}$P-labeled primers (top strand beginning 49 bp upstream or bottom strand beginning 62 bp downstream of the Fis core sites) and Vent (exo$^-$) DNA polymerase (New England Biolabs, Ipswich, MA, USA) (*Miller et al., 1996*). Primer extension products were precisely mapped on 8% polyacrylamide-7M urea gels, alongside sequencing ladders generated with the same labeled primers using the Sequenase Quick Denature Plasmid Sequencing Kit (USB-Affymetrix, Santa Clara, CA, USA).

## Invertasome modeling

Hin structural models of the DNA-bound dimers (based on PDB ID: 1GDT, *Yang and Steitz, 1995*), pre-cleaved tetramer (based on PDB ID: 3BVP, *Yuan et al., 2008*), and cleaved tetramer (based on PDB ID: 1ZR4, *Li et al., 2005*) combined with the Hin DBD-DNA structure (PDB ID: 1IJW) have been described previously (*Dhar et al., 2009a*, *2009b*; *Heiss et al., 2011*). The enhancer DNA model was generated using the DNA rebuild module in 3DNA (*Lu and Olson, 2003*). DNA parameter files were compiled from structures of the central 21 bp of the Fis-DNA co-crystal (PDB ID: 3IV5, *Stella et al., 2010*) for the two flanking Fis binding sites together with the native sequence for the intervening and flanking DNA generated by using mean DNA parameters from the protein-bound DNA library, which has an average helical twist value of 34.2° (*Olson et al., 1998*). The Fis dimers were then aligned onto their binding sites based on the co-crystal structure. The Hin DNA-cleaved and pre-cleaved tetramer models were manually docked onto the Fis-bound enhancer such that the Fis β-hairpin arms were optimally positioned with Arg154 and Leu155 on the Hin DBDs and consistent with the (−2) topology of a branched DNA on negatively supercoiled DNA. The model of Hin dimers associated with the enhancer was generated by independently docking Hin dimer-*hix* models onto the enhancer using the DNA-cleaved tetramer model as a guide to position the Hin DBD relative to Fis. Morphed intermediates (*Video 2*) were generated using the Yale Morphing Server (*Krebs and Gerstein, 2000*). All structural figures were generated in PyMOL (*DeLano, 2008*); surface electrostatic calculations utilized the APBS plug-in with a monovalent ion concentration of 0.15 M (*Baker et al., 2001*).

## Acknowledgements

We thank Feng Guo for comments on the manuscript.

## Additional information

### Funding

| Funder | Grant reference number | Author |
| --- | --- | --- |
| National Institute of General Medical Sciences | GM038509 | Meghan M McLean, Yong Chang, Gautam Dhar, John K Heiss, Reid C Johnson |
| National Institutes of Health, National Research Service Award | GM07104 | Meghan M McLean |

The funders had no role in study design, data collection and interpretation, or the decision to submit the work for publication.

### Author contributions

MMM, RCJ, Conception and design, Acquisition of data, Analysis and interpretation of data, Drafting or revising the article; YC, GD, JKH, Conception and design, Acquisition of data, Analysis and interpretation of data

# Additional files

## Major datasets

The following previously published datasets were used:

| Author(s) | Year | Dataset title | Dataset ID and/or URL | Database, license, and accessibility information |
|---|---|---|---|---|
| Li W, Kamtekar S, Xiong Y, Sarkis GJ, Grindley ND, Steitz TA | 2005 | Structure of a Synaptic gamma-delta Resolvase Tetramer Covalently linked to two Cleaved DNAs | http://www.rcsb.org/pdb/explore/explore.do?structureId=1zr4 | Publicly available at RCSB Protein Data Bank. |
| Chiu TK, Sohn C, Dickerson RE, Johnson RC | 2002 | Testing the Water-Mediated Hin Recombinase DNA Recognition by Systematic Mutations | http://www.rcsb.org/pdb/explore/explore.do?structureId=1ijw | Publicly available at RCSB Protein Data Bank. |
| Yuan P, Gupta K, Van Duyne GD | 2008 | Crystal Structure of the N-terminal Catalytic Domain of TP901-1 Integrase | http://www.rcsb.org/pdb/explore/explore.do?structureId=3bvp | Publicly available at RCSB Protein Data Bank. |
| Kamtekar S, Ho RS, Cocco MJ, Li W, Wenwieser SV, Boocock MR, Grindley ND, Steitz TA | 2006 | An activated, truncated gamma-delta resolvase tetramer | http://www.rcsb.org/pdb/explore/explore.do?structureId=2gm5 | Publicly available at RCSB Protein Data Bank. |
| Yang W, Steitz TA | 1995 | Crystal structure of the site-specific recombinase gamma delta resolvase complexed with a 34 bp cleavage site | http://www.rcsb.org/pdb/explore/explore.do?structureId=1gdt | Publicly available at RCSB Protein Data Bank. |
| Stella S, Cascio D, Johnson RC | 2010 | Crystal structure of Fis bound to 27 bp optimal binding sequence F1 | http://www.rcsb.org/pdb/explore/explore.do?structureId=3iv5 | Publicly available at RCSB Protein Data Bank. |

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
