## [Decision Letter]

Thank you for sending your work entitled “Multiple interfaces between a serine recombinase and an enhancer control site-specific DNA inversion” for consideration at *eLife*. Your article has been favorably evaluated by a Senior editor and 3 reviewers, one of whom, Leemor Joshua-Tor, is a member of our Board of Reviewing Editors.

The Reviewing editor and the other reviewers discussed their comments before we reached this decision, and the Reviewing editor has assembled the following comments to help you prepare a revised submission.

Thank you for submitting your manuscript to *eLife*. It has been carefully reviewed by three reviewers. All agreed that this is an important model system for both a general understanding of how multiprotein complexes assemble on DNA targets with well-separated binding sites and where supercoiling is involved, as well as more specifically for serine recombinase-family proteins. In addition, the paper presents a clever and comprehensive set of experiments that reveal the important contacts of the Hin invertasome both with Fis and with the enhancer DNA. The model for Hin/Fis assembly on DNA provides a satisfying explanation for how Fis stimulates inversion and how the system is wired to specifically carry out inversion with production of primarily unknotted product. A particularly appealing aspect of the work is that it makes use of existing structural models to carry out specific, targeted cross-linking experiments, along with targeted hydroxyl radical cleavage mapping, all of which are technically challenging. However, one of the most interesting findings, the unexpected similarity between invertases and resolvases, e.g., between Hin and Sin, in regions used to make interfaces in their respective synaptic complexes, could be highlighted more clearly in the manuscript.

The reviewers had several specific comments:

1) Role of the catalytic domain residues (47-51) in stimulating inversion (Figure 4, Figure 7, and related Discussion). The explanation of how Fis-Hin interactions could stabilize a synaptic tetramer of Hin subunits seems very plausible and is well supported by a lot of data. The explanation of how the catalytic domain residues contribute also seems reasonable, but one can't help but think that perhaps something else is going on as well. The interaction with enhancer DNA could help promote stable formation of the tetramer, but it is less clear why loss of these interactions leads to a higher frequency of multiple rotations and knotted products. The explanation given is that the Fis/enhancer can more easily detach, leading to less control of the reaction as the short loop increases in size. An additional/alternative source for an explanation can be considered from Figure 4, where the ‘unknotted recombinant’ is predicted to have DNA crossing over the catalytic domains in a way that does not exist in the parental configuration. Is it not possible that residues 47-51 could interact with this new DNA and contribute to product stabilization in a similar manner? This is more difficult to model than the enhancer DNA, but would be worth mentioning. On the other hand, local stiffness of the DNA might be enough to prevent it from making its way down to helix B even with a little extra torque as shown for the product. It might be useful to clarify your explanation of the topology experiment. Is it correct to say that, with the small loop, if the enhancer stays put, multiple rounds of recombination are energetically uphill because of the conformational strain they'd introduce in the dinky loop. If the enhancer wanders off, the loop can expand to accommodate additional nodes. The control experiment showing that loop size matters for the WT but not for the mutant is very satisfying in this regard.

2) Figure 4 describes nice experiments that show the helix B mutations are only needed if Fis is. The paper would be more interesting if it included some comparison with Rowland et al.'s quite similar experiments on Sin.

3) Discussion and Figure 8: the number of K and R near the end of helix B doesn't look significantly different in the invertases vs resolvases.

4) It should be discussed that the region of the C-terminus of helix B also includes the long-studied 2-3' γδ mutants (R52 and K54) and Sin mutants with analogous phenotypes (F52 and R54). Especially given that Hin, Sin, and γδ have otherwise-different synaptic complexes, and that this corner of the protein contacts other subunits in Sin and (presumably) γδ, but contacts DNA in Hin, it is reasonable to suspect that there is probably deeper significance to its recurrence in screens. Are the contacts made to the C-terminus of helix B only stabilizing the binding of other entities in the complex, or could they be some sort of “pull tab” by which the accessory proteins/DNA can actively destabilize the dimer and tip the conformational equilibrium toward the active tetramer? The sentence “Finally, attractive electrostatic...” suggests this idea, but better comparisons with γδ and Sin would bolster it nicely.

5) It isn’t evident that the mutants in the initial crosslinking experiments (K51 and K158 in particular) are folded properly, though this appears to be taken care of by Figure 4. Perhaps the H107Y controls could be more easily explained.

6) If one is unfamiliar with this system, it is sometimes hard to follow where some of the features, helix B etc. that are being discussed in various figures were positioned relative to the tetramer. Even some indication of which part of the structure was the DBD and which was the catalytic domain of Hin would be helpful, especially since this journal is aimed at a more general audience. How is the orientation in Figure 5 related to that in Figure 2? One suggestion is to show the rest of the complex as some sort of pale, transparent ghost.

In summary, there is substantial enthusiasm and we hope you would address the specific comments in a revised manuscript.

---

## [Author Response]

The editor's and reviewers’ comments (comments 2–4) suggested that we more fully relate and compare our results with Hin with those concerning the regulation and assembly of recombination complexes in related resolvase reactions. We agree and are happy that *eLife* allows the space to adequately address this. We have added an expanded section at the end of the Discussion that describes the important similarities between the systems and their general significance with respect to the control of serine recombinases. We highlight relevant regulatory residues in the Sin and γδ/Tn3 resolvase systems in Figure 8 and specifically state that resolvase hyperactive mutants containing regulatory mutations analogous to the Hin DBD (Sin) and helix-B (Sin and γδ/Tn3) interfaces are able to catalyze recombination in the absence of accessory sub-sites. We have added a figure (Figure 7—figure supplement 2) comparing the electrostatic surfaces of Hin and resolvase, which illustrates the differences in charge over the helix-B region as well as the alternative basic patch (part of the 2-3′ interface) on resolvase.

*1) Role of the catalytic domain residues (47-51) in stimulating inversion (*Figure 4*,*
Figure 7*, and related Discussion). The explanation of how Fis-Hin interactions could stabilize a synaptic tetramer of Hin subunits seems very plausible and is well supported by a lot of data. The explanation of how the catalytic domain residues contribute also seems reasonable, but one can't help but think that perhaps something else is going on as well. The interaction with enhancer DNA could help promote stable formation of the tetramer, but it is less clear why loss of these interactions leads to a higher frequency of multiple rotations and knotted products. The explanation given is that the Fis/enhancer can more easily detach, leading to less control of the reaction as the short loop increases in size. An additional/alternative source for an explanation can be considered from*
Figure 4*, where the ‘unknotted recombinant’ is predicted to have DNA crossing over the catalytic domains in a way that does not exist in the parental configuration. Is it not possible that residues 47-51 could interact with this new DNA and contribute to product stabilization in a similar manner? This is more difficult to model than the enhancer DNA, but would be worth mentioning. On the other hand, local stiffness of the DNA might be enough to prevent it from making its way down to helix B even with a little extra torque as shown for the product. It might be useful to clarify your explanation of the topology experiment. Is it correct to say that, with the small loop, if the enhancer stays put, multiple rounds of recombination are energetically uphill because of the conformational strain they'd introduce in the dinky loop. If the enhancer wanders off, the loop can expand to accommodate additional nodes. The control experiment showing that loop size matters for the WT but not for the mutant is very satisfying in this regard*.

The reviewers propose an additional/alternative model for why mutations in helix-B of the catalytic domain lead to processive DNA exchanges. If we understand correctly, the reviewers suggest that the DNA strand in the small loop is “captured” by the catalytic domains of the top subunits (as drawn in Figure 4) after DNA exchange, thereby inhibiting further exchanges. Although this is an intriguing idea, it seems to us to be unlikely. As illustrated in Figure 7, the green DNA emanating from the yellow Hin subunit would have to become severely kinked in order to associate with the helix-B region of the top subunits. As the reviewers note, local DNA stiffness, particularly if the DBD of the yellow subunit remains bound to DNA, would strongly disfavor such a configuration.

We have revised the section of the Discussion explaining the topological argument for why release of the enhancer (or the presence of long DNA segments between recombination sites and enhancer) can permit additional subunit rotations. We hope this revision plus small modifications in the Results help make the revised version clearer.

*4) It should be discussed that the region of the C-terminus of helix B also includes the long-studied 2-3' γδ mutants (R52 and K54) and Sin mutants with analogous phenotypes (F52 and R54). Especially given that Hin, Sin, and γδ have otherwise-different synaptic complexes, and that this corner of the protein contacts other subunits in Sin and (presumably) γδ, but contacts DNA in Hin, it is reasonable to suspect that there is probably deeper significance to its recurrence in screens. Are the contacts made to the C-terminus of helix B only stabilizing the binding of other entities in the complex, or could they be some sort of “pull tab” by which the accessory proteins/DNA can actively destabilize the dimer and tip the conformational equilibrium toward the active tetramer? The sentence “Finally, attractive electrostatic...” suggests this idea, but better comparisons with γδ and Sin would bolster it nicely*.

As noted in the top paragraph, most of this comment is now addressed in greater detail in the final section of the Discussion, along with Figure 8 and Figure 7—figure supplement 2. We certainly agree with the idea that helix-B likely functions to “pull” the tetramer to its active conformation. To give this important point greater emphasis we have inserted “…driven by coulombic forces between the helix-B region and the enhancer DNA…” in the paragraph above the sentence referred to by the reviewers: “Finally, attractive electrostatic forces between the helix-B region and the enhancer DNA promote the final conformational change into the active tetramer structure.” We discuss this again in the final section at the end (both in the Hin summary and in relation to the resolvase systems).

*5) It isn’t evident that the mutants in the initial crosslinking experiments (K51 and K158 in particular) are folded properly, though this appears to be taken care of by*
Figure 4*. Perhaps the H107Y controls could be more easily explained*.

The reviewers are correct that the experiments in Figures 2 and 4 effectively rule out a folding problem by alanine substitutions at the critical residues in the DBD helix-1 and helix-B. This has now been specifically stated. While we cannot rule out a small effect on folding in the case of the Lys51 and Lys158 mutants, the amount of covalent Hin-DNA cleavage products in the crosslinking experiments (Figure 2), the BMOE crosslinking with K158C (Figure 2,) and the Hin inversion activities of K51A and K158A (Table 1), argue that at least a substantial fraction of these mutant proteins are correctly folded. Nevertheless, we have qualified the last sentence of the second paragraph of Results to say “These results provide an initial indication…”

*6) If one is unfamiliar with this system, it is sometimes hard to follow where some of the features, helix B etc. that are being discussed in various figures were positioned relative to the tetramer. Even some indication of which part of the structure was the DBD and which was the catalytic domain of Hin would be helpful, especially since this journal is aimed at a more general audience. How is the orientation in*
Figure 5
*related to that in*
Figure 2*? One suggestion is to show the rest of the complex as some sort of pale, transparent ghost*.

We have labeled the DBD and catalytic domains on the Hin structure in Figure 2 and helix-B in Figure 4. We prepared a version of Figure 5 containing the rest of the complex as a transparent ghost. However, we feel this makes the figure much more complex, preempts Figure 7, and de-emphasizes the major point of the figure, which is the locations of the FeBABE cleavage sites. We now state in the figure legend that the Hin molecules are rotated 40° about the y-axis relative to Figure 2 and refer to Figure 7 showing the whole model. A figure supplement including the rest of the complex as a transparent ghost could be linked to Figure 5 if it is felt to be necessary.